# Discriminator optimal transport

**Akinori Tanaka**
Mathematical Science Team, RIKEN Center for Advanced Intelligence Project (AIP)
1-4-1 Nihonbashi, Chuo-ku, Tokyo 103-0027, Japan
Interdisciplinary Theoretical and Mathematical Sciences Program (iTHEMS), RIKEN
2-1 Hirosawa, Wako, Saitama 351-0198, Japan
Department of Mathematics, Faculty of Science and Technology, Keio University
3-14-1 Hiyoshi, Kouhoku-ku, Yokohama 223-8522, Japan
`akinori.tanaka@riken.jp`

## Abstract

Within a broad class of generative adversarial networks, we show that discriminator optimization process increases a lower bound of the dual cost function for the Wasserstein distance between the target distribution $p$ and the generator distribution $p_G$. It implies that the trained discriminator can approximate optimal transport (OT) from $p_G$ to $p$. Based on some experiments and a bit of OT theory, we propose discriminator optimal transport (DOT) scheme to improve generated images. We show that it improves inception score and FID calculated by unconditional GAN trained by CIFAR-10, STL-10 and a public pre-trained model of conditional GAN trained by ImageNet.

## 1 Introduction

Generative Adversarial Network (GAN) [1] is one of recent promising generative models. In this context, we prepare two networks, a generator $G$ and a discriminator $D$. $G$ generates fake samples $G(z)$ from noise $z$ and tries to fool $D$. $D$ classifies real sample $x$ and fake samples $y = G(z)$. In the training phase, we update them alternatingly until it reaches to an equilibrium state. In general, however, the training process is unstable and requires tuning of hyperparameters. Since from the first successful implementation by convolutional neural nets [2], most literatures concentrate on *how to improve the unstable optimization procedures* including changing objective functions [3, 4, 5, 6, 7, 8], adding penalty terms [9, 10, 11], techniques on optimization precesses themselves [12, 13, 14, 15], inserting new layers to the network [16, 17], and others we cannot list here completely.

Even if one can make the optimization relatively stable and succeed in getting $G$ around an equilibrium, it sometimes fails to generate meaningful images. Bad images may include some unwanted structures like unnecessary shadows, strange symbols, and blurred edges of objects. For example, see generated images surrounded by blue lines in Figure 1. These problems may be fixed by scaling up the network structure and the optimization process. Generically speaking, however, it needs large scale computational resources, and if one wants to apply GAN to individual tasks by making use of more compact devices, the above problem looks inevitable and crucial.

There is another problem. In many cases, we discard the trained discriminator $D$ after the training. This situation is in contrast to other latent space generative models. For example, variational auto-encoder (VAE) [18] is also composed of two distinct networks, an encoder network and a decoder network. We can utilize both of them after the training: the encoder can be used as a data compressor, and the decoder can be regarded as a generator. Compared to this situation, it sounds wasteful to use only $G$ after the GAN training.

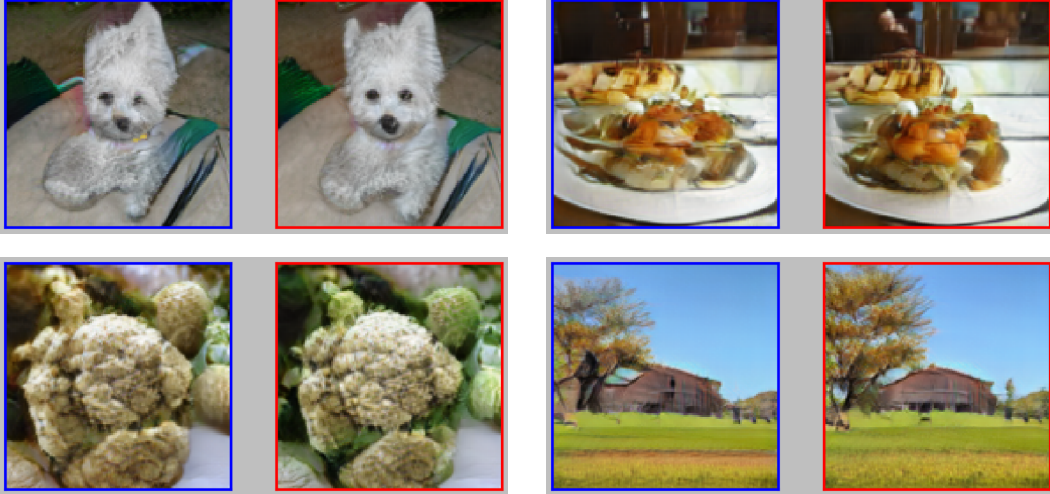

Figure 1: Each left image (blue): a sample from generator $G$. Each right image (red): the sample modified by our algorithm based on discriminator $D$. We use here the trained model available on `https://github.com/pfnet-research/sngan_projection` .

From this viewpoint, it would be natural to ask *how to use trained models $G$ and $D$ efficiently*. Recent related works in the same spirit are discriminator rejection sampling (DRS) [19] and Metropolis-Hastings GAN (MH-GAN) [20]. In each case, they use the generator-induced distribution $p_G$ as a proposal distribution, and approximate acceptance ratio of the proposed sample based on the trained $D$. Intuitively, generated image $\boldsymbol{y} = G(\boldsymbol{z})$ is accepted if the value $D(\boldsymbol{y})$ is relatively large, otherwise it is rejected. They show its theoretical backgrounds, and it actually improve scores on generated images in practice.

In this paper, we try a similar but more active approach, i.e. improving generated image $\boldsymbol{y} = G(\boldsymbol{z})$ directly by adding $\delta\boldsymbol{y}$ to $\boldsymbol{y}$ such that $D(\boldsymbol{y} + \delta\boldsymbol{y}) > D(\boldsymbol{y})$. The optimal transport (OT) theory guarantees that the improved samples can be regarded as approximate samples from the target distribution $p$. More concretely, our contributions are

- Proposal of the discriminator optimal transport (DOT) based on the fact that the objective function for $D$ provides lower bound of the dual cost function for the Wasserstein distance between $p$ and $p_G$.

- Suggesting approximated algorithms and verifying that they improve Earth Mover's distance (EMD) [21], inception score [13] and Fréchet inception distance (FID) [15].

- Pointing out a *generality* on DOT, i.e. if the $G$'s output domain is same as the $D$'s input domain, then we can use *any* pair of trained $G$ and $D$ to improve generated samples.

In addition, we show some results on experiment comparing DOT and a naive method of improving sample just by increasing the value of $D$, under a fair setting. One can download our codes from `https://github.com/AkinoriTanaka-phys/DOT`.

## 2 Background

### 2.1 Generative Adversarial Nets

Throughout this paper, we regard an image sample as a vector in certain Euclidean space: $\boldsymbol{x} \in X$. We name latent space as $Z$ and a prior distribution on it as $p_Z(\boldsymbol{z})$. The ultimate goal of the GAN is making generator $G : Z \to X$ whose push-foward of the prior $p_G(\boldsymbol{x}) = \int_Z p_Z(\boldsymbol{z})\delta\big(\boldsymbol{x} - G(\boldsymbol{z})\big)d\boldsymbol{z}$ reproduces data-generating probability density $p(\boldsymbol{x})$. To achieve it, discriminator $D : X \to \mathbb{R}$ and

objective functions,

$$V_D(G, D) = \mathbb{E}_{\boldsymbol{x} \sim p} \left[ f(-D(\boldsymbol{x})) \right] + \mathbb{E}_{\boldsymbol{y} \sim p_G} \left[ f(D(\boldsymbol{y})) \right], \tag{1}$$

$$V_G(G, D) = \mathbb{E}_{\boldsymbol{y} \sim p_G} \left[ g(D(\boldsymbol{y})) \right], \tag{2}$$

are introduced. Choice of functions $f$ and $g$ corresponds to choice of GAN update algorithm as explained below. Practically, $G$ and $D$ are parametric models $G_\theta$ and $D_\varphi$, and they are alternatingly updated as

$$\varphi \leftarrow \varphi + \epsilon \nabla_\varphi V_D(G_\theta, D_\varphi), \tag{3}$$

$$\theta \leftarrow \theta - \epsilon \nabla_\theta V_G(G_\theta, D_\varphi), \tag{4}$$

until the updating dynamics reaches to an equilibrium. One of well know choices for $f$ and $g$ is

$$f(u) = -\log(1 + e^u) \quad g(u) = -f(-u). \tag{5}$$

Theoretically speaking, it seems better to take $g(u) = f(u)$, which is called minimax GAN [22] to guarantee $p_G = p$ at the equilibrium as proved in [1]. However, it is well known that taking (5), called non-saturating GAN, enjoys better performance practically. As an alternative, we can choose the following $f$ and $g$ [6, 4]:

$$f(u) = \max(0, -1 - u), \quad g(u) = -u. \tag{6}$$

It is also known to be relatively stable. In addition to it, $p_G = p$ at an equilibrium is proved at least in the theoretically ideal situation. Another famous choice is taking

$$f(u) = -u, \quad g(u) = u. \tag{7}$$

The resultant GAN is called WGAN [5]. We use (7) with gradient penalty (WGAN-GP) [9] in our experiment. WGAN is related to the concept of the optimal transport (OT) which we review below, so one might think our method is available only when we use WGAN. But we would like to emphasize that such OT approach is also useful even when we take GANs described by (5) and (6) as we will show later.

## 2.2 Spectral normalization

Spectral normalization (SN) [16] is one of standard normalizations on neural network weights to stabilize training process of GANs. To explain it, let us define a norm for function called Lipschitz norm,

$$||f||_{Lip} := \sup \left\{ \frac{||f(\boldsymbol{x}) - f(\boldsymbol{y})||_2}{||\boldsymbol{x} - \boldsymbol{y}||_2} \Big| \boldsymbol{x} \neq \boldsymbol{y} \right\}. \tag{8}$$

For example, $||ReLU||_{Lip} = ||lReLU||_{Lip} = 1$ because their maximum gradient is 1. For linear transformation $l_{W,b}$ with weight matrix $W$ and bias $b$, the norm $||l_{W,b}||_{Lip}$ is equal to the maximum singular value $\sigma(W)$. Spectral normalization on $l_{W,b}$ is defined by dividing the weight $W$ in the linear transform by the $\sigma(W)$:

$$SN(l_{W,b}) = l_{W/\sigma(W),b}. \tag{9}$$

By definition, it enjoys $||l_{W/\sigma(W)}||_{Lip} = 1$. If we focus on neural networks, estimation of the upper bound of the norm is relatively easy because of the following property[1]:

$$||f \circ g||_{Lip} \leq ||f||_{Lip} \cdot ||g||_{Lip}. \tag{10}$$

For example, suppose $f_{nn}$ is a neural network with ReLU or lReLU activations and spectral normalizations: $f_{nn}(\boldsymbol{x}) = SN \circ l_{W_L} \circ f \circ SN \circ l_{W_{L-1}} \circ \cdots \circ SN \circ l_{W_1}(\boldsymbol{x})$, then the Lipschitz norm is bounded by one:

$$||f_{nn}||_{Lip} \leq \prod_{l=1}^{L} ||l_{W_l/\sigma(W_l)}||_{Lip} = 1 \tag{11}$$

Thanks to this Lipschitz nature, the normalized network gradient behaves mild during repeating updates (3) and (4), and as a result, it stabilizes the wild and dynamic optimization process of GANs.

## 2.3 Optimal transport

Another important background in this paper is optimal transport. Suppose there are two probability densities, $p(\boldsymbol{x})$ and $q(\boldsymbol{y})$ where $\boldsymbol{x}, \boldsymbol{y} \in X$. Let us consider the cost for transporting one unit of mass from $\boldsymbol{x} \sim p$ to $\boldsymbol{y} \sim q$. The optimal cost is called Wasserstein distance. Throughout this paper, we focus on the Wasserstein distance defined by $l_2$-norm cost $||\boldsymbol{x} - \boldsymbol{y}||_2$:

$$W(p, q) = \min_{\pi \in \Pi(p,q)} \left( \mathbb{E}_{(\boldsymbol{x}, \boldsymbol{y}) \sim \pi} \Big[ ||\boldsymbol{x} - \boldsymbol{y}||_2 \Big] \right). \tag{12}$$

$\pi$ means joint probability for transportation between $\boldsymbol{x}$ and $\boldsymbol{y}$. To realize it, we need to restrict $\pi$ satisfying marginality conditions,

$$\int d\boldsymbol{x} \, \pi(\boldsymbol{x}, \boldsymbol{y}) = q(\boldsymbol{y}), \quad \int d\boldsymbol{y} \, \pi(\boldsymbol{x}, \boldsymbol{y}) = p(\boldsymbol{x}). \tag{13}$$

An optimal $\pi^*$ satisfies $W(p, q) = \mathbb{E}_{(\boldsymbol{x}, \boldsymbol{y}) \sim \pi^*}[||\boldsymbol{x} - \boldsymbol{y}||_2]$, and it realizes the most effective transport between two probability densities under the $l_2$ cost. Interestingly, (12) has the dual form

$$W(p, q) = \max_{||\tilde{D}||_{Lip} \leq 1} \left( \mathbb{E}_{\boldsymbol{x} \sim p} \Big[ \tilde{D}(\boldsymbol{x}) \Big] - \mathbb{E}_{\boldsymbol{y} \sim q} \Big[ \tilde{D}(\boldsymbol{y}) \Big] \right). \tag{14}$$

The duality is called Kantorovich-Rubinstein duality [23, 24]. Note that $||f||_{Lip}$ is defined in (8), and the dual variable $\tilde{D}$ should satisfy Lipschitz continuity condition $||\tilde{D}||_{Lip} \leq 1$. One may wonder whether any relationship between the optimal transport plan $\pi^*$ and the optimal dual variable $D^*$ exists or not. The following theorem is an answer and it plays a key role in this paper.

**Theorem 1** *Suppose $\pi^*$ and $D^*$ are optimal solutions of the primal* (12) *and the dual* (14) *problem, respectively. If $\pi^*$ is deterministic optimal transport described by a certain automorphism[2] $T :$ $X \to X$, then the following equations are satisfied:*

$$||D^*||_{Lip} = 1, \tag{15}$$

$$T(\boldsymbol{y}) = \arg\min_{\boldsymbol{x}} \left\{ ||\boldsymbol{x} - \boldsymbol{y}||_2 - D^*(\boldsymbol{x}) \right\}, \tag{16}$$

$$p(\boldsymbol{x}) = \int d\boldsymbol{y} \, \delta\Big( \boldsymbol{x} - T(\boldsymbol{y}) \Big) q(\boldsymbol{y}). \tag{17}$$

(Proof) It can be proved by combining well know facts. See Supplementary Materials. □

## 3 Discriminator optimal transport

If we apply the spectral normalization on a discriminator $D$, it satisfies $||D||_{Lip} = K$ with a certain real number $K$. By redefining it to $\tilde{D} = D/K$, it becomes 1-Lipschitz $||\tilde{D}||_{Lip} = 1$. It reminds us the equation (15), and one may expect a connection between OT and GAN. In fact, we can show the following theorem:

**Theorem 2** *Each objective function of GAN using logistic* (5)*, or hinge* (6)*, or identity* (7) *loss with gradient penalty, provides lower bound of the mean discrepancy of $\tilde{D} = D/K$ between $p$ and $p_G$:*

$$V_D(G, D) \leq K \left( \mathbb{E}_{\boldsymbol{x} \sim p} \Big[ \tilde{D}(\boldsymbol{x}) \Big] - \mathbb{E}_{\boldsymbol{y} \sim p_G} \Big[ \tilde{D}(\boldsymbol{y}) \Big] \right). \tag{18}$$

(Proof) See Supplementary Materials. □

In practical optimization process of GAN, $K$ could change its value during the training process, but it stays almost constant at least approximately as explained below.

$$\min_{T: X \to X} \left( \mathbb{E}_{\boldsymbol{y} \sim q} \Big[ ||T(\boldsymbol{y}) - \boldsymbol{y}||_2 \Big] \right), \quad \text{constrained by (17).}$$

Reconstructing $T^*$ from $\pi^*$ without any assumption is a subtle problem and only guaranteed within strictly convex cost functions [25]. Unfortunately, it is not satisfied in our $l_2$ cost. However, there is a known method [26] to find a solution based on relaxing the cost to strictly convex cost $||\boldsymbol{x} - \boldsymbol{y}||_2^{1+\epsilon}$ with $\epsilon > 0$. In our experiments, DOT works only when $||\boldsymbol{x} - \boldsymbol{y}||_2$ is small enough for given $\boldsymbol{y}$. In this case, there is no big difference between $||\boldsymbol{x} - \boldsymbol{y}||_2$ and $||\boldsymbol{x} - \boldsymbol{y}||_2^{1+\epsilon}$, and it suggests DOT approximates their solution.

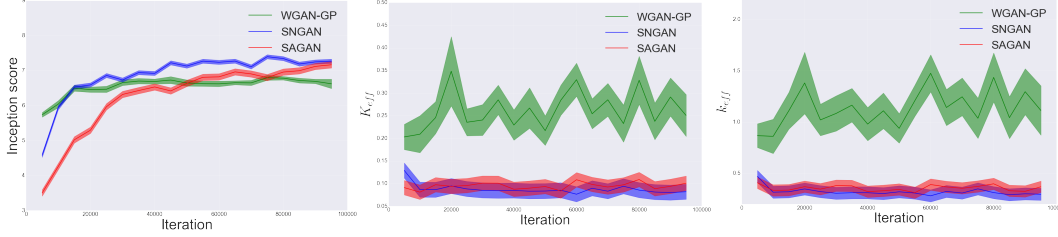

Figure 2: Logs of inception score (left), approximated Lipschitz constant of $D$ (middle), and approximated Lipschitz constant of $D \circ G$ (right) on each GAN trained with CIFAR-10. Approximated Lipschitz constants are calculated by random 500 pair samples. Errorbars are plotted within $1\sigma$ by 500 trials.

### 3.1 Discriminator Optimal Transport (ideal version)

The inequality (18) implies that the update (3) of $D$ during GANs training maximizes the lower bound of the objective in (14), the dual form of the Wasserstein distance. In this sense, the optimization of $D$ in (3) can be regarded solving the problem (14) approximately[3]. If we apply (16) with $D^* \approx \tilde{D} = D/K$, the following transport of given $\boldsymbol{y} \sim p_G$

$$T_D(\boldsymbol{y}) = \arg\min_{\boldsymbol{x}} \left\{ ||\boldsymbol{x} - \boldsymbol{y}||_2 - \frac{1}{K} D(\boldsymbol{x}) \right\} \tag{19}$$

is expected to recover the sampling from the target distribution $p$ thanks to the equality (17).

### 3.2 Discriminator Optimal Transport (practical version)

To check whether $K$ changes drastically or not during the GAN updates, we calculate approximated Lipschitz constants defined by

$$K_{\text{eff}} = \max\left\{ \frac{|D(\boldsymbol{x}) - D(\boldsymbol{y})|}{||\boldsymbol{x} - \boldsymbol{y}||_2} \Big| \boldsymbol{x}, \boldsymbol{y} \sim p_G \right\}, \tag{20}$$

$$k_{\text{eff}} = \max\left\{ \frac{|D \circ G(\boldsymbol{z}) - D \circ G(\boldsymbol{z_y})|}{||\boldsymbol{z} - \boldsymbol{z_y}||_2} \Big| \boldsymbol{z}, \boldsymbol{z_y} \sim p_Z \right\}, \tag{21}$$

in each 5,000 iteration on GAN training with CIFAR-10 data with DCGAN models explained in Supplementary Materials. As plotted in Figure 2, both of them do not increase drastically. It is worth to mention that the naive upper bound of the Lipschitz constant like (11) turn to be overestimated. For example, SNGAN has the naive upper bound 1, but (20) stays around 0.08 in Figure 2.

**Target space DOT**　Based on these facts, we conclude that trained discriminators can approximate the optimal transport (16) by

$$T_D^{\text{eff}}(\boldsymbol{y}) = \arg\min_{\boldsymbol{x}} \left\{ ||\boldsymbol{x} - \boldsymbol{y}||_2 - \frac{1}{K_{\text{eff}}} D(\boldsymbol{x}) \right\}. \tag{22}$$

As a preliminary experiment, we apply DOT to WGAN-GP trained by 25gaussians dataset and swissroll dataset. We use the gradient descent method shown in Algorithm 1 to search transported point $T_D^{\text{eff}}(\boldsymbol{y})$ for given $\boldsymbol{y} \sim p_G$. In Figure 3, we compare the DOT samples and naively transported samples by the discriminator which is implemented by replacing the gradient in Algorithm 1 to $-\frac{1}{K_{\text{eff}}} \nabla_{\boldsymbol{x}} D(\boldsymbol{x})$, i.e. just searching $\boldsymbol{x}$ with large $D(\boldsymbol{x})$ from initial condition $\boldsymbol{x} \leftarrow \boldsymbol{y}$ where $\boldsymbol{y} \sim p_G$.

DOT outperforms the naive method qualitatively and quantitatively. On the 25gaussians, one might think 4th naively improved samples are better than 3rd DOT samples. However, the 4th samples are too concentrated and lack the variance around each peak. In fact, the value of the Earth Mover's distance, EMD, which measures how long it is separated from the real samples, shows relatively large value. On the swissroll, 4th samples based on naive transport lack many relevant points close to the original data, and it is trivially bad. On the other hand, one can see that the 3rd DOT samples keep swissroll shape and clean the blurred shape in the original samples by generator.

**Algorithm 1** Target space optimal transport by gradient descent

---

**Require:** trained $D$, approximated $K_{\text{eff}}$ by (20), sample $\boldsymbol{y}$, learning rate $\epsilon$ and small vector $\boldsymbol{\delta}$

    Initialize $\boldsymbol{x} \leftarrow \boldsymbol{y}$

    **for** $n_{\text{trial}}$ in range($N_{\text{updates}}$) **do**

        $\boldsymbol{x} \leftarrow \boldsymbol{x} - \epsilon \boldsymbol{\nabla}_{\boldsymbol{x}} \left\{ ||\boldsymbol{x} - \boldsymbol{y} + \boldsymbol{\delta}||_2 - \frac{1}{K_{\text{eff}}} D(\boldsymbol{x}) \right\}$    ( $\boldsymbol{\delta}$ is for preventing overflow. )

    **end for**

    **return** $\boldsymbol{x}$

---

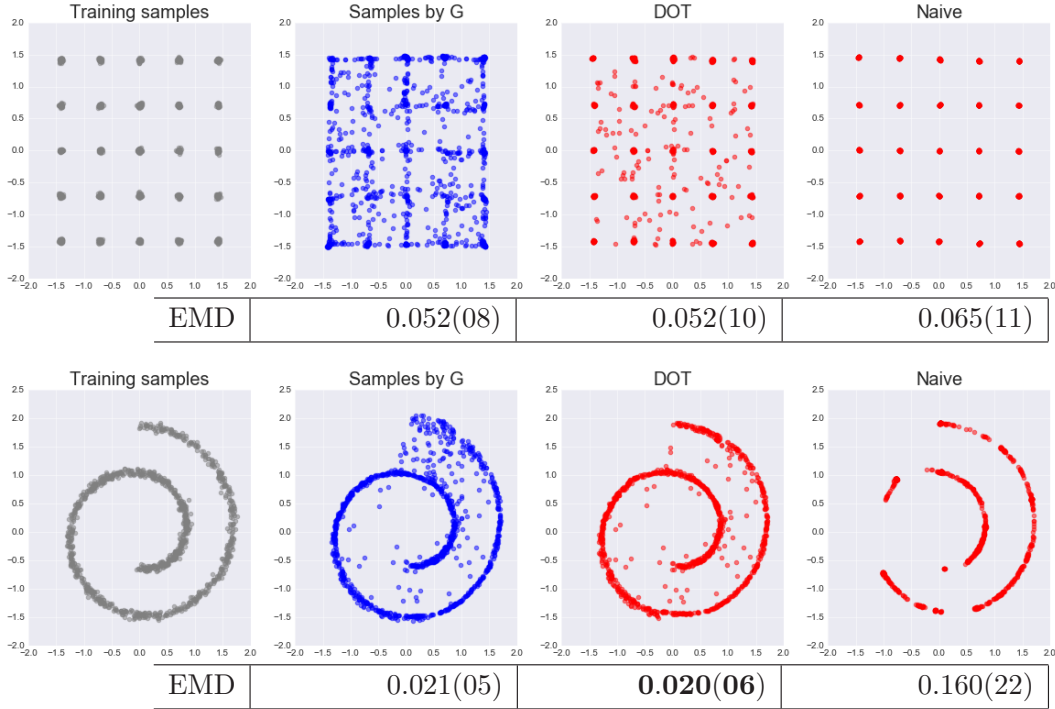

Figure 3: 2d experiments by using trained model of WGAN-GP. 1,000 samples of, 1st: training samples, 2nd: generated samples by $G$, 3rd: samples by target space DOT, 4th: samples by naive transport, are plotted. Each EMD value is calculated by 100 trials. The error corresponds to $1\sigma$. We use $\boldsymbol{\delta} = 0.001$. See the supplementary material for more details on this experiment.

**Latent space DOT** The target space DOT works in low dimensional data, but it turns out to be useless once we apply it to higher dimensional data. See Figure 4 for example. Alternative, and more workable idea is regarding $D \circ G : Z \to \mathbb{R}$ as the dual variable for defining Wasserstein distance between "pullback" of $p$ by $G$ and prior $p_Z$. Latent space OT itself is not a novel idea [27, 28], but there seems to be no literature using trained $G$ and $D$, to the best of our knowledge. The approximated Lipschitz constants of $G \circ D$ also stay constant as shown in the right sub-figure in Figure 2, so we conclude that

$$T_{D \circ G}^{\text{eff}}(\boldsymbol{z_y}) = \underset{\boldsymbol{z}}{\arg\min} \left\{ ||\boldsymbol{z} - \boldsymbol{z_y}||_2 - \frac{1}{k_{\text{eff}}} D \circ G(\boldsymbol{z}) \right\} \tag{23}$$

approximates optimal transport in latent space. Note that if the prior $p_Z$ has non-trivial support, we need to restrict $\boldsymbol{z}$ onto the support during the DOT process. In our algorithm 2, we apply projection of the gradient. One of the major practical priors is normal distribution $\mathcal{N}(0, \boldsymbol{I}_{D \times D})$ where $D$ is the latent space dimension. If $D$ is large, it is well known that the support is concentrated on $(D-1)$-dimensional sphere with radius $\sqrt{D}$, so the projection of the gradient $\boldsymbol{g}$ can be calculated by $\boldsymbol{g} - (\boldsymbol{g} \cdot \boldsymbol{z})\boldsymbol{z}/\sqrt{D}$ approximately. Even if we skip this procedure, transported images may look improved, but it downgrades inception scores and FIDs.

**Algorithm 2** Latent space optimal transport by gradient descent
___
**Require:** trained $G$ and $D$, approximated $k_{\text{eff}}$, sample $z_y$, learning rate $\epsilon$, and small vector $\delta$
  Initialize $z \leftarrow z_y$
  **for** $n_{\text{trial}}$ in range($N_{\text{updates}}$) **do**
    $g = \nabla_z \left\{ ||z - z_y + \delta||_2 - \frac{1}{k_{\text{eff}}} D \circ G(z) \right\}$    ( $\delta$ is for preventing overflow. )
    **if** noise is generated by $\mathcal{N}(0, I_{D \times D})$ **then**
      $g \leftarrow g - (g \cdot z)z/\sqrt{D}$
    **end if**
    $z \leftarrow z - \epsilon g$
    **if** noise is generated by $U([-1, 1])$ **then**
      clip $z \in [-1, 1]$
    **end if**
  **end for**
  **return** $x = G(z)$
___

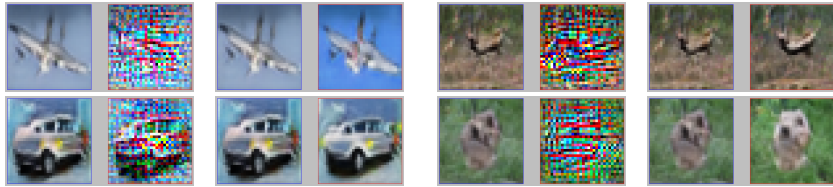

Figure 4: Target space DOT sample (each left) and latent space DOT sample (each right). The former looks like giving meaningless noises like perturbations in adversarial examples [29]. On the other hand, the latent space DOT samples keep the shape of image, and clean it.

## 4 Experiments on latent space DOT

### 4.1 CIFAR-10 and SLT-10

We prepare pre-trained DCGAN models and ResNet models on various settings, and apply latent space DOT. In each case, inception score and FID are improved (Table 1). We can use arbitrary discriminator $D$ to improve scores by fixed $G$ as shown in Table 2. As one can see, DOT really works. But it needs tuning of hyperparameters. First, it is recommended to use small $\epsilon$ as possible. A large $\epsilon$ may accelerate upgrading, but easily downgrade unless appropriate $N_{\text{updates}}$ is chosen. Second, we recommend to use $k_{\text{eff}}$ calculated by using enough number of samples. If not, it becomes relatively small and it also possibly downgrade images. As a shortcut, $k_{\text{eff}} = 1$ also works. See Supplementary Materials for details and additional results including comparison to other methods.

| | | CIFAR-10 | | STL-10 | |
|---|---|---|---|---|---|
| | | bare | DOT | bare | DOT |
| DCGAN | WGAN-GP | 6.53(08), 27.84 | 7.45(05), 24.14 | 8.69(07), 49.94 | 9.31(07), 44.45 |
| | SNGAN(ns) | 7.45(09), 20.74 | 7.97(14), **15.78** | 8.67(01), 41.18 | 9.45(13), **34.84** |
| | SNGAN(hi) | 7.45(08), 20.47 | 8.02(16), 17.12 | 8.83(12), 40.10 | 9.35(12), 34.85 |
| | SAGAN(ns) | 7.75(07), 25.37 | **8.50(01)**, 20.57 | 8.68(01), 48.23 | 10.04(14), 41.19 |
| | SAGAN(hi) | 7.52(06), 25.78 | 8.38(05), 21.21 | 9.29(13), 45.79 | **10.30(21)**, 40.51 |
| Resnet | SAGAN(ns) | 7.74(09), 22.13 | 8.49(13), 20.22 | 9.33(08), 41.91 | **10.03(14), 39.48** |
| | SAGAN(hi) | 7.85(11), 21.53 | **8.50(12), 19.71** | | |

Table 1: (Inception score, FID) by usual sampling (bare) and DOT: Models in [16] and self-attention layer [17] are used. (ns) and (hi) mean models trained by (5) and (6). $\epsilon = 0.01$ SGD is applied 20 times for CIDAR-10 and 10 times for STL-10. $k_{\text{eff}}$ is calculated by 100 samples and $\delta = 0.001$.

| $D$ | without $D$ | WGAN-gp | SNGAN(ns) | SNGAN(hi) | SAGAN(ns) | SAGAN(hi) |
|---|---|---|---|---|---|---|
| IS | 7.52(06) | 8.03(11) | 8.22(07) | 8.38(07) | 8.36(12) | 8.38(05) |
| FID | 25.78 | 24.47 | 21.45 | 23.03 | 21.07 | 21.21 |

Table 2: Results on scores by $G_{\text{SAGAN(ns)}}$ after latent space DOT using each $D$ in different training scheme using CIFAR-10 within DCGAN architecture. Parameters for DOT are same in Table 1.

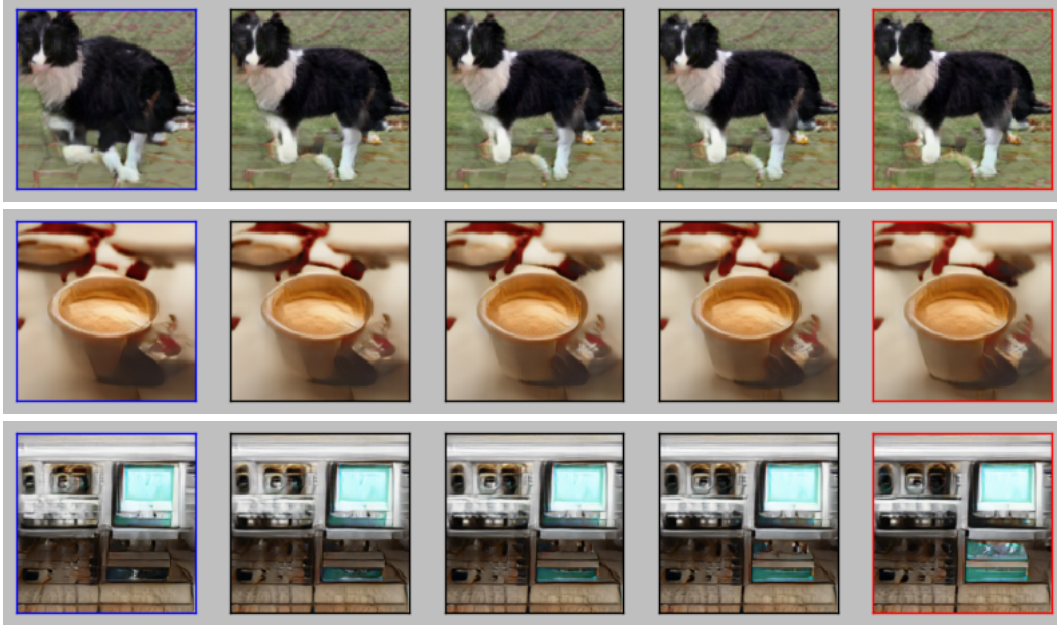

Figure 5: Left images surrounded by blue lines are samples from the conditional generator. The number of updates $N_{\text{updates}}$ for DOT increases along horizontal axis. Right Images surrounded by red lines corresponds after 30 times updates with Adam $(\alpha, \beta_1, \beta_2) = (0.01, 0, 0.9)$ and $k_{\text{eff}}(y) = 1$.

## 4.2 ImageNet

**Conditional version of latent space DOT**    In this section, we show results on ImageNet dataset. As pre-trained models, we utilize a pair of public models $(G, D)$ [30] of conditional GAN [31][4]. In conditional GAN, $G$ and $D$ are networks conditioned by label $y$. Typical objective function $V_D$ is therefore represented by average over the label:

$$V_D(G, D) = \mathbb{E}_{y \sim p(y)} \Big[ V_D \Big( G(\cdot|y), D(\cdot|y) \Big) \Big]. \tag{24}$$

But, once $y$ is fixed, $G(\boldsymbol{z}|y)$ and $D(\boldsymbol{x}|y)$ can be regarded as usual networks with input $\boldsymbol{z}$ and $\boldsymbol{x}$ respectively. So, by repeating our argument so far, DOT in conditional GAN can be written by

$$T_{G \circ D}(\boldsymbol{z_y}|y) = \operatorname{argmin}_{\boldsymbol{z}} \Big\{ ||\boldsymbol{z} - \boldsymbol{z_y}||_2 - \frac{1}{k_{\text{eff}}(y)} D\Big( G(\boldsymbol{z}|y)|y \Big) \Big\}. \tag{25}$$

where $k_{\text{eff}}(y)$ is approximated Lipschitz constant conditioned by $y$. It is calculated by

$$k_{\text{eff}}(y) = \max \Big\{ \frac{|D\big(G(\boldsymbol{z}|y)|y\big) - D\big(G(\boldsymbol{z_y}|y)|y\big)|}{||\boldsymbol{z} - \boldsymbol{z_y}||_2} \Big| \boldsymbol{z}, \boldsymbol{z_y} \sim p_Z \Big\}. \tag{26}$$

**Experiments**    We apply gradient descent updates with with Adam$(\alpha, \beta_1, \beta_2) = (0.01, 0, 0.9)$. We show results on 4 independent trials in Table 3. It is clear that DOT mildly improve each score. Note that we need some tunings on hyperparameters $\epsilon$, $N_{\text{updates}}$ as we already commented in 4.1.

|  | # updates=0 | # updates=4 | # updates=16 | # updates=32 |
|---|---|---|---|---|
| trial1($k_{\text{eff}}(y) = 1$) | 36.40(91), 43.34 | 36.99(75), 43.01 | 37.25(84), 42.70 | **37.61(88), 42.35** |
| trial2($k_{\text{eff}}(y) = 1$) | 36.68(59), 43.60 | 36.26(98), 43.09 | 36.97(63), 42.85 | **37.02(73), 42.74** |
| trial3 | 36.64(63), 43.55 | 36.87(84), 43.11 | **37.51(01), 42.43** | 36.88(79), 42.52 |
| trial4 | 36.23(98), 43.63 | 36.49(54), 43.25 | 37.29(86), 42.67 | **37.29(07), 42.40** |

Table 3: (Inception score, FID) for each update. Upper 2 cases are executed by $k_{\text{eff}}(y) = 1$ without calculating (26). We use 50 samples for each label $y$ to calculate $k_{\text{eff}}(y)$ in lower 2 trials. $\boldsymbol{\delta = 0.001}$.

**Algorithm 3** Latent space conditional optimal transport by gradient descent

---

**Require:** trained $G$ and $D$, label $y$, approximated $k_{\text{eff}}(y)$, sample $\boldsymbol{z_y}$, learning rate $\epsilon$ and small vector $\boldsymbol{\delta}$

    Initialize $\boldsymbol{z} \leftarrow \boldsymbol{z_y}$

    **for** $n_{\text{trial}}$ in range($N_{\text{update}}$) **do**

        $\boldsymbol{g} = \boldsymbol{\nabla_z}\left\{ ||\boldsymbol{z} - \boldsymbol{z_y} + \boldsymbol{\delta}||_2 - \frac{1}{k_{\text{eff}}(y)} D\Big(G(\boldsymbol{z}|y)\Big|y\Big) \right\}$      ( $\boldsymbol{\delta}$ is for preventing overflow. )

        **if** noise is generated by $\mathcal{N}(0, \boldsymbol{I}_{D \times D})$ **then**

            $\boldsymbol{g} \leftarrow \boldsymbol{g} - (\boldsymbol{g} \cdot \boldsymbol{z})\boldsymbol{z}/\sqrt{D}$

        **end if**

        $\boldsymbol{z} \leftarrow \boldsymbol{z} - \epsilon \boldsymbol{g}$

        **if** noise is generated by $U([-1, 1])$ **then**

            clip $\boldsymbol{z} \in [-1, 1]$

        **end if**

    **end for**

    **return** $\boldsymbol{x} = G(\boldsymbol{z}|y)$

---

**Evaluation** To calculate FID, we use available 798,900 image files in ILSVRC2012 dataset. We reshape each image to the size $299 \times 299 \times 3$, feed all images to the public inception model to get the mean vector $\boldsymbol{m}_w$ and the covariance matrix $\boldsymbol{C}_w$ in 2,048 dimensional feature space explained in Supplementary Materials.

## 5 Conclusion

In this paper, we show the relevance of discriminator optimal transport (DOT) method on various trained GAN models to improve generated samples. Let us conclude with some comments here.

First, DOT objective function in (22) reminds us the objective for making adversarial examples [29]. There is known fast algorithm to make adversarial example making use of the piecewise-linear structure of the ReLU neural network [32]. The method would be also useful for accelerating DOT.

Second, latent space DOT can be regarded as improving the prior $p_Z$. A similar idea can be found also in [33]. In the usual context of the GAN, we fix the prior, but it may be possible to train the prior itself simultaneously by making use of the DOT techniques.

We leave these as future works.

### Acknowledgments

We would like to thank Asuka Takatsu for fruitful discussion and Kenichi Bannai for careful reading this manuscript. This work was supported by computational resources provided by RIKEN AIP deep learning environment (RAIDEN) and RIKEN iTHEMS.

## Footnotes

[1] This inequality can be understood as follows. Naively, the norm (8) is defined by the maximum gradient between two different points. Suppose $\boldsymbol{x}_1$ and $\boldsymbol{x}_2$ realizing maximum of gradient for $g$ and $\boldsymbol{u}_1$ and $\boldsymbol{u}_2$ are points for $f$, then the (RHS) of the inequality (10) is equal to $||f(\boldsymbol{u}_1) - f(\boldsymbol{u}_2)||_2/||\boldsymbol{u}_1 - \boldsymbol{u}_2||_2 \times ||g(\boldsymbol{x}_1) - g(\boldsymbol{x}_2)||_2/||\boldsymbol{x}_1 - \boldsymbol{x}_2||_2$. If $g(\boldsymbol{x}_i) = \boldsymbol{u}_i$, it reduces to the (LHS) of the (10), but the condition is not satisfied in general, and the (RHS) takes a larger value than (LHS). This observation is actually important to the later part of this paper because estimation of the norm based on the inequality seems to be overestimated in many cases.

[2] It is equivalent to assume there exists an unique solution of the corresponding Monge problem:

[3] This situation is similar to guarantee VAE [18] objective function which is a lower bound of the likelihood called evidence lower bound (ELBO).

[4] These are available on `https://github.com/pfnet-research/sngan_projection` .

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
