[Supplementary Material]

# Supplementary Materials of Paper #3691

**Akinori Tanaka**
Mathematical Science Team, RIKEN Center for Advanced Intelligence Project (AIP)
1-4-1 Nihonbashi, Chuo-ku, Tokyo 103-0027, Japan
Interdisciplinary Theoretical and Mathematical Sciences Program (iTHEMS), RIKEN
2-1 Hirosawa, Wako, Saitama 351-0198, Japan
Department of Mathematics, Faculty of Science and Technology, Keio University
3-14-1 Hiyoshi, Kouhoku-ku, Yokohama 223-8522, Japan
`akinori.tanaka@riken.jp`

## Abstract

We provide proofs of theorems used in the main body of the paper first. After that, we show experimental details and some additional experimental results.

## A  Proofs

### A.1  Proof of Theorem 1

We show a proof of **Theorem 1** here by utilizing well known propositions in optimal transport [1, 2]. First, we show the following proposition for later use.

**Proposition 1** *Suppose $\pi^*$ and $D^*$ are optimal solutions of primal and dual problem respectively, then the equation*

$$\int d\boldsymbol{x} d\boldsymbol{y}\ \pi^*(\boldsymbol{x}, \boldsymbol{y}) \Big[ ||\boldsymbol{x} - \boldsymbol{y}||_2 - \Big( D^*(\boldsymbol{x}) - D^*(\boldsymbol{y}) \Big) \Big] = 0 \tag{1}$$

*is satisfied.*

(Proof) Thanks to the strong duality, we have

$$\mathbb{E}_{(\boldsymbol{x}, \boldsymbol{y}) \sim \pi^*} \Big[ ||\boldsymbol{x} - \boldsymbol{y}||_2 \Big] = \mathbb{E}_{\boldsymbol{x} \sim p} \Big[ D^*(\boldsymbol{x}) \Big] - \mathbb{E}_{\boldsymbol{y} \sim p_G} \Big[ D^*(\boldsymbol{y}) \Big]. \tag{2}$$

Now, let us remind that $\pi^*(\boldsymbol{x}, \boldsymbol{y})$ satisfies the marginality conditions $p(\boldsymbol{x}) = \int d\boldsymbol{y}\ \pi^*(\boldsymbol{x}, \boldsymbol{y})$ and $q(\boldsymbol{y}) = \int d\boldsymbol{y}\ \pi^*(\boldsymbol{x}, \boldsymbol{y})$. It means we can replace the (RHS) of (2) by

$$\mathbb{E}_{(\boldsymbol{x}, \boldsymbol{y}) \sim \pi^*} \Big[ D^*(\boldsymbol{x}) - D^*(\boldsymbol{y}) \Big]. \tag{3}$$

By transposing it to (LHS) of (2), it completes the proof.□

As a corollary of the proposition, we can show the first identity in **Theorem 1**,

$$||D^*||_{Lip} = \sup \Big\{ \frac{|D^*(\boldsymbol{x}) - D^*(\boldsymbol{y})|}{||\boldsymbol{x} - \boldsymbol{y}||_2} \Big| \boldsymbol{x} \neq \boldsymbol{y} \Big\} = 1. \tag{4}$$

First, let us remind that $||D^*||_{Lip} \leq 1$ is automatically satisfied. It means for arbitrary $\boldsymbol{x}$ and $\boldsymbol{y}$,

$$\Big[ ||\boldsymbol{x} - \boldsymbol{y}||_2 - |D^*(\boldsymbol{x}) - D^*(\boldsymbol{y})| \Big] \geq 0 \tag{5}$$

is satisfied. Next, $-x \geq -|x|$ is trivially true for arbitrary $x \in \mathbb{R}$. By using this inequality with $x = D^*(\boldsymbol{x}) - D^*(\boldsymbol{y})$, we conclude

$$\left[ ||\boldsymbol{x} - \boldsymbol{y}||_2 - \left( D^*(\boldsymbol{x}) - D^*(\boldsymbol{y}) \right) \right] \geq 0. \tag{6}$$

It means the integrand in the equation (1) is always positive or zero. Then, we can say

$$\pi^*(\boldsymbol{x}, \boldsymbol{y}) \neq 0 \quad \Rightarrow \quad \left[ ||\boldsymbol{x} - \boldsymbol{y}||_2 - \left( D^*(\boldsymbol{x}) - D^*(\boldsymbol{y}) \right) \right] = 0, \tag{7}$$

because if not, we cannot cancel its contribution in the integral (1). $\pi^*$ is probability density, so there exists a pair $(\boldsymbol{x}, \boldsymbol{y})$ satisfying $\pi^*(\boldsymbol{x}, \boldsymbol{y}) \neq 0$, and the pair realizes the absolute gradient 1. As already noted, $D^*$ should satisfy $||D^*||_{Lip} \leq 1$, so (7) means there exists two element $\boldsymbol{x}$ and $\boldsymbol{y}$ realizing this upper bound, i.e. $||D^*||_{Lip} = 1$.

The second equation

$$T(\boldsymbol{y}) = \arg\min_{\boldsymbol{x}} \left\{ ||\boldsymbol{x} - \boldsymbol{y}||_2 - D^*(\boldsymbol{x}) \right\}, \tag{8}$$

is also proved as a corollary of **Proposition 1**. But we need to use a help of the assumption in **Theorem 1**, i.e. the existence of the deterministic solution $T$ of the Monge's problem. It means $\pi^*(\boldsymbol{x}, \boldsymbol{y})$ is deterministic by a certain automorphism $T$ and described by Dirac's delta function[1] with respect to $\boldsymbol{x}$ for given $\boldsymbol{y}$, i.e.

$$\begin{aligned} \pi^*(\boldsymbol{x}, \boldsymbol{y}) \neq 0 &\quad \text{if } \boldsymbol{x} = T(\boldsymbol{y}), \\ \pi^*(\boldsymbol{x}, \boldsymbol{y}) = 0 &\quad \text{otherwise.} \end{aligned} \tag{9}$$

Because of $||D^*||_{Lip} = 1$,

$$-D^*(\boldsymbol{y}) \leq ||\boldsymbol{x} - \boldsymbol{y}||_2 - D^*(\boldsymbol{x}) \tag{10}$$

is satisfied for arbitrary $\boldsymbol{x}$. On the other hand, thanks to the equality (7),

$$-D^*(\boldsymbol{y}) = ||T(\boldsymbol{y}) - \boldsymbol{y}||_2 - D^*(T(\boldsymbol{y})) \tag{11}$$

should be satisfied. It means the (RHS) of (11) is the minimum value of (RHS) of (10), and it completes the proof of (8).

The third identity

$$p(\boldsymbol{x}) = \int d\boldsymbol{y} \, \delta\left( \boldsymbol{x} - T(\boldsymbol{y}) \right) q(\boldsymbol{y}), \tag{12}$$

can be got as a corollary of the following proposition.

**Proposition 2** *If $\pi^*$ is the deterministic solution of the primal problem, then it should be represented by*

$$\pi^*(\boldsymbol{x}, \boldsymbol{y}) = \delta\left( \boldsymbol{x} - T(\boldsymbol{y}) \right) q(\boldsymbol{y}) \tag{13}$$

*with the optimal transport map $T : X \to X$.*

(Proof) First of all, because of the assumption (9), $\pi^*$ should be proportional to $\delta(\boldsymbol{x} - T(\boldsymbol{y}))$. To satisfy the marginal conditions of $\pi$, we multiply a function of $\boldsymbol{x}, \boldsymbol{y}$ to it:

$$\pi^*(\boldsymbol{x}, \boldsymbol{y}) = \delta\left( \boldsymbol{x} - T(\boldsymbol{y}) \right) f(\boldsymbol{x}, \boldsymbol{y}). \tag{14}$$

Thanks to the delta function, however, it is sufficient to take into account $f(T^{-1}(\boldsymbol{y}), \boldsymbol{y})$ and let us call it $g(\boldsymbol{y})$, then

$$\pi^*(\boldsymbol{x}, \boldsymbol{y}) = \delta\left( \boldsymbol{x} - T(\boldsymbol{y}) \right) g(\boldsymbol{y}). \tag{15}$$

Now, let us consider the marginal condition on $\boldsymbol{y}$, i.e. integration over $\boldsymbol{x}$ should be equal to $q(\boldsymbol{y})$:

$$q(\boldsymbol{y}) = \int d\boldsymbol{x}\, \pi^*(\boldsymbol{x}, \boldsymbol{y}) = \int d\boldsymbol{x}\, \delta\Big(\boldsymbol{x} - T(\boldsymbol{y})\Big) g(\boldsymbol{y}) = g(\boldsymbol{y}). \tag{16}$$

It completes the proof.$\square$

In the above proof, we do not consider taking marginal along $\boldsymbol{y}$ which gives (12) by definition. One may be suspicious on it. In fact, by directly integrating it over $\boldsymbol{y}$, we get

$$\int d\boldsymbol{y}\, \delta\Big(\boldsymbol{x} - T(\boldsymbol{y})\Big) q(\boldsymbol{y}) = \frac{q(\boldsymbol{y})}{|\det \nabla_{\boldsymbol{y}} T(\boldsymbol{y})|}\Big|_{\boldsymbol{y} = T^{-1}(\boldsymbol{x})}. \tag{17}$$

But it is known that the (RHS) actually agree with $p(\boldsymbol{x})$. Physical meaning of this fact is simple. Now, let $T^{-1} : X \to X$ is the optimal transportation from $p(\boldsymbol{x})$ to $q(\boldsymbol{y})$. The numerator of (17) is just a map of mass of the probability density, and the denominator corresponds to the Jacobian to guarantee its integration over $\boldsymbol{x}$ is 1.

$$\int d\boldsymbol{x}\, \frac{q(\boldsymbol{y})}{|\det \nabla_{\boldsymbol{y}} T(\boldsymbol{y})|}\Big|_{\boldsymbol{y} = T^{-1}(\boldsymbol{x})} = \int d[T(\boldsymbol{y})] \frac{q(\boldsymbol{y})}{|\det \nabla_{\boldsymbol{y}} T(\boldsymbol{y})|} = \int d\boldsymbol{y}\, q(\boldsymbol{y}) = 1. \tag{18}$$

For more detail, see the chapter 11 in [1] for example.

### A.2 Proof of Theorem 2

It is sufficient to show

$$V_D(G, D) \le \mathbb{E}_{\boldsymbol{x} \sim p}\Big[D(\boldsymbol{x})\Big] - \mathbb{E}_{\boldsymbol{y} \sim p_G}\Big[D(\boldsymbol{y})\Big] \tag{19}$$

because the $\tilde{D}(\boldsymbol{x})$ is defined by $\tilde{D}(\boldsymbol{x}) = D(\boldsymbol{x})/K$. Below, we show this inequality in each case.

**Logistic**  Because of the monotonicity of log,

$$-\log(1 + e^a) \le -\log e^a \tag{20}$$

is satisfied for arbitrary $a \in \mathbb{R}$. So the objective $V_D$ defined by logistic loss enjoys

$$\begin{aligned}
V_D(G, D) &= -\mathbb{E}_{\boldsymbol{x} \sim p}\Big[\log\Big(1 + e^{-D(\boldsymbol{x})}\Big)\Big] - \mathbb{E}_{\boldsymbol{y} \sim p_G}\Big[\log\Big(1 + e^{+D(\boldsymbol{y})}\Big)\Big] \\
&\le -\mathbb{E}_{\boldsymbol{x} \sim p}\Big[\log\Big(e^{-D(\boldsymbol{x})}\Big)\Big] - \mathbb{E}_{\boldsymbol{y} \sim p_G}\Big[\log\Big(e^{+D(\boldsymbol{y})}\Big)\Big] \\
&= \mathbb{E}_{\boldsymbol{x} \sim p}\Big[D(\boldsymbol{x})\Big] - \mathbb{E}_{\boldsymbol{y} \sim p_G}\Big[D(\boldsymbol{y})\Big].
\end{aligned} \tag{21}$$

**Hinge**  On the hinge loss, we use the inequality

$$\min(0, u) \le u \tag{22}$$

as follows.

$$\begin{aligned}
V_D(G, D) &= \mathbb{E}_{\boldsymbol{x} \sim p}\Big[\min\Big(0, -1 + D(\boldsymbol{x})\Big)\Big] + \mathbb{E}_{\boldsymbol{y} \sim p_G}\Big[\min\Big(0, -1 - D(\boldsymbol{y})\Big)\Big] \\
&\le \mathbb{E}_{\boldsymbol{x} \sim p}\Big[-1 + D(\boldsymbol{x})\Big] + \mathbb{E}_{\boldsymbol{y} \sim p_G}\Big[-1 - D(\boldsymbol{y})\Big] \\
&\le \mathbb{E}_{\boldsymbol{x} \sim p}\Big[D(\boldsymbol{x})\Big] - \mathbb{E}_{\boldsymbol{y} \sim p_G}\Big[D(\boldsymbol{y})\Big].
\end{aligned} \tag{23}$$

**Gradient penalty**  The objective function for discriminator in WGAN-GP is

$$V_D(G, D) = \mathbb{E}_{\boldsymbol{x} \sim p}\Big[D(\boldsymbol{x})\Big] - \mathbb{E}_{\boldsymbol{y} \sim p_G}\Big[D(\boldsymbol{y})\Big] - \text{penalty}, \tag{24}$$

and the penalty term is defined by

$$\text{penalty} = \lambda \mathbb{E}_{\boldsymbol{x}}\Big[\big|\big|\nabla_{\boldsymbol{x}} D(\boldsymbol{x}) - 1\big|\big|^2\Big] \ge 0 \tag{25}$$

for a certain positive value $\lambda$ which immediately gives the inequality.

# B  Details on experiments

## B.1  2d experiment

**Training of GAN**   We use same artificial data used in [4]. 25 gaussians data is generated as follows. First, we generate 100,000 samples from $\mathcal{N}(\mathbf{0},(1\text{e-}2)\cdot\boldsymbol{I}_{2\times2})$. After that, we divide samples to 25 classes of 4,000 sub-samples and rearrange their center to $\{-4,-2,0,+2,+4\}\times\{-4,-2,0,+2,+4\}\subset\mathbb{R}^2$. To make the data variance 1, we divide all sample coordinates by 2.828. Swissroll data is generated by scikit-learn with 100,000 samples with noise=0.25. The swissroll data coordinates are also divided by 7.5.

We only use WGAN-GP in this experiment. The number of update for $D$ is 100 if number of iteration is less than 25 and 10 otherwise per one update for $G$. We apply Adam with $(\alpha,\beta_1,\beta_2)=(1\text{e-}4,0.5,0.9)$ to both of $G$ and $D$. Under these setup, we train WGAN-GP 20k times with batchsize 256. We summarize the structure of our models in Table 1.

| $z\sim U([-1,1]^2)$ |
| --- |
| dense $\rightarrow$ 256 lReLU |
| dense $\rightarrow$ 256 lReLU |
| dense $\rightarrow$ 256 lReLU |
| dense $\rightarrow$ 2 |
| (i) Generator |

| 2d vector $x\in\mathbb{R}^2$ |
| --- |
| dense $\rightarrow$ 512 lReLU |
| dense $\rightarrow$ 512 lReLU |
| dense $\rightarrow$ 512 lReLU |
| dense $\rightarrow$ 1 |
| (ii) Discriminator |

Table 1:  GAN architecture in 2d experiment.

**DOT**   First, we calculate the $K_{\text{eff}}$. We draw 100 pairs of independent samples $(\boldsymbol{x},\boldsymbol{y})$ from $p_G$ for calculating their gradient by $l_2$-norm, and take the maximum gradient as $K_{\text{eff}}$. In the experiment, we run 10 independent trials and take mean value. Actual values of $K_{\text{eff}}$ are 1.68 for 25 gaussians and 1.34 for swissroll in the experiment.

To apply the target space DOT shown in Algorithm 1, we use Adam optimizer with $(\alpha,\beta_1,\beta_2)=(0.01,0,0.9)$ for searching $\boldsymbol{x}$ in both of DOT and Naive transports. We run the gradient descent 100 times, and calculate the Earth-Mover's distance (EMD) between randomly chosen 1,000 training samples and 1,000 generated samples by each method. We repeat this procedure 100 times, and get the mean value and std of the EMD.

**EMD**   Earth Mover's distance (EMD) can be regarded as a discrete version of the Wasserstein distance. Suppose $\{x_i\}_{i=1,2,\ldots,N}$ and $\{y_i\}_{i=1,2,\ldots,N}$ are samples on $X$. EMD is defined by

$$\text{EMD}(\{x_i\},\{y_j\})=\min_{\pi\in\Pi(\{x_i\},\{y_j\})}\sum_{i,j=1}^{N}\pi_{ij}d(x_i,y_j),\tag{26}$$

where $\pi$ is constraint on

$$\pi_{ij}\in\{0,\frac{1}{N}\},\quad\sum_{i=1}^{N}\pi_{ij}=\sum_{j=1}^{N}\pi_{ij}=\frac{1}{N}.\tag{27}$$

If we regard samples as discrete approximation of the distribution, EMD measures how two distributions are separated. So if $x_i$ and $y_j$ are sampled from same distribution, the value is expected to be close to zero. In our paper, we use python library [5] to calculate it. It used $d(x,y)=||x-y||_2^2$ by default.

## B.2  Experiments on CIFAR-10 and STL-10

**Training of GAN**   We use conventional CIFAR-10 dataset. On STL-10, we downsize it to $48\times48$ instead of using the original size $96\times96$. Each pixel is normalized so that it takes value in $[-1,1]$.

On WGAN and SNGAN, we apply 5 updates for $D$ per 1 update of $G$ and use Adam on $G$ and $D$ with same hyperparameter: $(\alpha, \beta_1, \beta_2) = (0.0002, 0.0, 0.9)$. We use the gradient penalty on WGAN with $\lambda = 10.0$. On SAGAN, we apply "two timescale update rule" [6], i.e. 1 update for $D$ per 1 update of $G$, and Adam with $(\alpha^G, \beta_1^G, \beta_2^G) = (0.0001, 0.0, 0.9)$ and $(\alpha^D, \beta_1^D, \beta_2^D) = (0.0004, 0.0, 0.9)$. Under this setting, we update each GAN 150k times with batchsize 64, except for ResNet SAGAN on STL-10 which is trained by 240k times in the same setup.

We use conventional DCGAN with or without self-attention (SA) layer and normalized layers by spectral normalization (SN) and ResNet including SA layer. We use usual DCGAN architecture on WGAN and SNGAN used in [7]. On SAGAN, we insert a self-attention layer on the layer with 128 channels because it enjoys the best performance within our trials. We show our DCGAN model in Table 2 and ResNet in Table 3.

| $\boldsymbol{z} \sim U([-1,1]^{128})$ |
| :---: |
| (SN)dense $\to$ BN $\to M_g \times M_g \times 512$ |
| $4 \times 4$, str=2, pad=1, (SN)deconv. BN 256 ReLU |
| $4 \times 4$, str=2, pad=1, (SN)deconv. BN 128 ReLU |
| (128 SA with 16 channels) |
| $4 \times 4$, str=2, pad=1, (SN)deconv. BN 64 ReLU |
| $3 \times 3$, str=1, pad=1, (SN)deconv. 3 Tanh |
| (i) Generator |

| RGB image $\boldsymbol{x} \in [-1,1]^{M \times M \times 3}$ |
| :---: |
| $3 \times 3$, str=1, pad=1, (SN)conv 64 lReLU |
| $4 \times 4$, str=2, pad=1, (SN)conv 128 lReLU |
| (128 SA with 16 channels) |
| $3 \times 3$, str=1, pad=1, (SN)conv 128 lReLU |
| $4 \times 4$, str=2, pad=1, (SN)conv 256 lReLU |
| $3 \times 3$, str=1, pad=1, (SN)conv 256 lReLU |
| $4 \times 4$, str=2, pad=1, (SN)conv 512 lReLU |
| $3 \times 3$, str=1, pad=1, (SN)conv 512 lReLU |
| (SN)dense $\to$ 1 |
| (ii) Discriminator |

Table 2: DCGAN model. In WGAN, we use bare dense, deconv, conv without self-attentions in both generator and discriminator. In SNGAN, we use SNdense and SNconv in discriminator but bare dense and deconv in generator without self-attentions. In SAGAN, all SN and SA are turned on. We use the SA layer defined in Figure 1. We use $(M_g, M) = (4, 32)$ in CIFAR-10, $(M_g, M) = (6, 48)$ in STL-10.

Figure 1: $c$ self-attention with $s$ channels.

| $z \sim N(0, \boldsymbol{I}_{128n \times 128n})$ |
| --- |
| SNdense → ReLU $M_g \times M_g \times 128n$ |
| SNResBlock up 128n |
| SNResBlock up 128n |
| SNResBlock up 128n |
| 128n SA with16n channels |
| ReLU, BN, $3 \times 3$ (SN)conv, 3 Tanh |

(i) Generator

| RGB image $\boldsymbol{x} \in [-1, 1]^{M \times M \times 3}$ |
| --- |
| SNResBlock down1 128 |
| SNResBlock down2 128 |
| SNResBlock down3 128 |
| SNResBlock down3 128 |
| 128 SA with 16 channels |
| ReLU, SNdense → 1 |

(ii) Discriminator

Table 3: ResNet model. In this paper, we concentrate on ResNet with spectral normalization and self attention trained by CIFAR-10 (n, $M_g$, M)=(2, 4, 32), SLT-10(n, $M_g$, M)=(1, 6, 48). See Figure 2 and Figure 3 for definitions of ResBlocks.

**DOT** First of all, let us pay attention to the implementation of SN proposed in [7]. The algorithm gradually approximate SN by Monte Carlo sampling based on forward propagations, and does not give well normalized weights in the beginning, so we should be careful to apply DOT on such network. One easy way is just running forward propagations a few times. Before each DOT, we run forward propagation on $G$ and $D$ to thermalize the SN layers. We apply SGD update with[2] lr=0.01. In Table 1 of the main paper, we update each generated samples with 20 times for DCGAN, 10 times for ResNet.

To get $k_{\text{eff}}$, we draw 100 pairs of samples $(\boldsymbol{x}, \boldsymbol{y})_i$, calculate maximum gradient, and define it as $k_{\text{eff}}$. However, there seems no big difference to use $k_{\text{eff}}$ in high precision or not. To compare them, we executed $k_{\text{eff}} = 1$ DOT and summarize scores (IS, FID) on 0, 10 and 20 updates.

| | # updates $= 0$ | # updates $= 10$ | # updates $= 20$ |
| --- | --- | --- | --- |
| trial1($k_{\text{eff}} = 1.00$) | $6.47(05), 27.83$ | $7.17(07), 24.31$ | $7.35(01), 24.06$ |
| trial2($k_{\text{eff}} = 0.86$) | $6.53(08), 27.84$ | $7.21(01), 24.06$ | $7.45(05), 24.14$ |

WGAN-GP(CIFAR-10, lr=0.01)

| Number of updates | # updates $= 0$ | # updates $= 10$ | # updates $= 20$ |
| --- | --- | --- | --- |
| trial1($k_{\text{eff}} = 1.00$) | $7.44(01), 20.71$ | $7.63(05), 18.38$ | $7.69(09), 17.74$ |
| trial2($k_{\text{eff}} = 0.39$) | $7.45(09), 20.74$ | $7.85(08), 16.57$ | $7.97(14), 15.78$ |

SNGAN(ns)(CIFAR-10, lr=0.01)

| Number of updates | # updates $= 0$ | # updates $= 10$ | # updates $= 20$ |
| --- | --- | --- | --- |
| trial1($k_{\text{eff}} = 1.00$) | $7.4(01), 20.32$ | $7.6(07), 19.3$ | $7.61(08), 19.01$ |
| trial2($k_{\text{eff}} = 0.34$) | $7.45(08), 20.47$ | $7.81(08), 17.72$ | $8.02(16), 17.12$ |

SNGAN(hi)(CIFAR-10, lr=0.01)

| Number of updates | # updates $= 0$ | # updates $= 10$ | # updates $= 20$ |
| --- | --- | --- | --- |
| trial1($k_{\text{eff}} = 1.00$) | $7.66(07), 25.09$ | $7.92(14), 23.1$ | $8.02(12), 22.48$ |
| trial2($k_{\text{eff}} = 0.28$) | $7.75(07), 25.37$ | $8.35(11), 21.27$ | $8.5(01), 20.57$ |

SAGAN(ns)(CIFAR-10, lr=0.01)

| Number of updates | # updates $= 0$ | # updates $= 10$ | # updates $= 20$ |
| --- | --- | --- | --- |
| trial1($k_{\text{eff}} = 1.00$) | $7.46(01), 26.08$ | $7.75(11), 24.12$ | $7.87(09), 23.33$ |
| trial2($k_{\text{eff}} = 0.21$) | $7.52(06), 25.78$ | $8.2(08), 21.45$ | $8.38(05), 21.21$ |

SAGAN(hi)(CIFAR-10, lr=0.01)

As one can see, lower $k_{\text{eff}}$ makes improvement faster. But please note that if it is too small, the DOT may be equivalent just decreasing $-D(\boldsymbol{x})$, and easily increase FID.

Figure 2: ResBlocks

Figure 3: ResBlocks

On the lr of gradient decent, it is better to take small value as possible. For example, the history of DOT for ResNet on STL-10 is as follows.

|  | # updates = 0 | # updates = 10 | # updates = 20 |
|---|---|---|---|
| Inception score | 9.33(08) | 10.03(14) | 10.00(12) |
| FID | 41.91 | 39.48 | 40.53 |

In this model we use $\mathcal{N}(0, \boldsymbol{I}_{128 \times 128})$ as the prior $p_Z$ and apply the projection of the gradient to conduct DOT updates. But our projection update is an approximation, and the slightly bad scores on 20 updates may be caused by $z$ getting out of the support because of too large lr. On the other hand, our DCGAN model has $U[-1, 1]^{128}$ as the prior, and there is no need of the projection. In this case, for example, SAGAN(hi)'s history is

|  | # updates = 0 | # updates = 10 | # updates = 20 |
|---|---|---|---|
| Inception score | 9.29(13) | 10.11(14) | 10.29(21) |
| FID | 45.78 | 41.09 | 40.51 |

and each score improved even after 10 updates. We show some results on DOT histories with different lr in Figure 4, Figure 5, Figure 6 also. As we can see from Figure 4, larger lr makes improvement faster, e.g. IS 7.40 reaches 8.88 and FID 22.37 reaches 17.61 at 10 update point, but it easily makes bad scores when we increase the number of updates, e.g. IS 8.88 at 10 reaches 7.79 and FID 17.61 at 10 reaches 58.64 at 90 update point. This is resolved by taking lower lr, but too low lr makes improvement too slow as we can see in Figure 6.

**Inception score and FID**   Inception score is defined by

$$IS(\{\boldsymbol{x}_i\}_i) = \exp \sum_{i=1}^{N} \frac{1}{N}\Big(\hat{D}(p(\boldsymbol{d}|\boldsymbol{x}_i)\|p(\boldsymbol{d}))\Big), \tag{28}$$

where $p(\boldsymbol{d}|\boldsymbol{x})$ is the output values of the inception model, and $p(\boldsymbol{d})$ is marginal distribution of $p(\boldsymbol{d}|\boldsymbol{x})p_G(\boldsymbol{x})$. This is one of well know metrics on GAN, and measures how the images $\{\boldsymbol{x}_i\}_i$ look realistic and how the images have variety. Usually, higher value is better.

The second well know metric is the Fréchet inception distance (FID). This value is the Wasserstein-2 distance between dataset and $\{\boldsymbol{x}_i\}_{i=1,2,\dots,N}$ in the 2,048 dimensional feature space of the inception model by assuming the distribution is gaussian. To compute it, we prepare the 2,048 dimensional mean vector $\boldsymbol{m}_w$ and covariant matrix $\boldsymbol{C}_w$ of the corresponding dataset, and calculate $\boldsymbol{m}$ and $\boldsymbol{C}$ by feeding $\{\boldsymbol{x}_i\}_{i=1,2,\dots,N}$ to the inception model. Then, the FID is calculated by

$$\text{FID}(\{\boldsymbol{x}_i\}_i) = \|\boldsymbol{m} - \boldsymbol{m}_w\|_2^2 + \text{Tr}(\boldsymbol{C} + \boldsymbol{C}_w - 2(\boldsymbol{C}\boldsymbol{C}_w)^{1/2}). \tag{29}$$

Note that the square root of the matrix is taken under matrix product, not component-wise root as usually taken in numpy. Lower FID is better.

**DOT vs Naive**   Here, we compare the latent space DOT and the latent space Naive improvement:

$$T_{D\circ G}^{\text{naive}}(\boldsymbol{z}_{\boldsymbol{y}}) = \text{argmin}_{\boldsymbol{z}}\Big\{-\frac{1}{k_{\text{eff}}}D\circ G(\boldsymbol{z})\Big\}. \tag{30}$$

As one can see from Figure 5 and Figure 6, both the DOT and the naive transport (30) improve scores. In Figure 5, DOT and Naive keep improving the inception score, on the other hand, the FID seems saturated around 40~50 updates. After that, one can see both of transports do not improve FID. Even worse, FID starts to increase both cases at some point of updates. Compared to the naive update, however, DOT can suppress it, but increasing FID at some update point seems inevitable. So, keeping lr low value as possible seems important as we have already noted.

**DOT vs MH-GAN**   There are some methods of post-processing using trained models of GAN [8, 9]. In this section, we focus on the Metropolis-Hastings GAN (MH-GAN) [9] which is relatively easy to implement. In MH-GAN, we first calibrate the trained discriminator by logistic regression, and use it as approximator of the accept/reject probability in the context of the Markov-Chain Monte-Carlo method for sampling. We calibrate $D$ by using $1,000$ training data and $1,000$ generated data, and run MC update $500$ times.

| | CIFAR-10 | | STL-10 | |
|---|---|---|---|---|
| | bare | MH-GAN | bare | MH-GAN |
| WGAN-GP | 6.5(08), 27.93 | 7.23(11), 36.14 | 8.71(13), 49.98 | **8.98(13), 48.03** |
| SNGAN(ns) | 7.42(09), 20.73 | 7.16(01), 23.24 | 8.62(15), 41.35 | 8.0(11), 46.27 |
| SNGAN(hi) | 7.44(08), 20.53 | **8.23(12), 18.57** | 8.78(01), 40.11 | **10.02(08), 36.34** |
| SAGAN(ns) | 7.69(08), 24.97 | **7.87(07), 22.48** | 8.63(08), 48.33 | **9.79(12), 44.44** |
| SAGAN(hi) | 7.52(06), 25.77 | **7.92(09), 23.75** | 9.32(11), 45.66 | 9.73(19), 49.1 |

Table 4: (Inception score, FID) by usual sampling (bare) and MH-GAN in within our DCGAN models. The bold letter scores correspond increasing inception score and decreasing FID.

We succeed in improving almost all inception scores except for SNGAN(ns) cases. On FID, however, MH-GAN sometimes downgrade it (taking higher value compared to its original value). By comparing Table 4 and Table 1 in the main body of this paper, DOT looks better in all cases, but we do not insist our method outperform MH-GAN here because our DOT method needs tuning parameters $\epsilon$, $k_{\text{eff}}$ besides tuning the number of update.

## B.3   On runtimes

We just used gradient of $G$ and $D$, so it scales same as the backprop. For reference, we put down real runtimes (seconds/30updates) here by Tesla P100:

| Swissroll | CIFAR-10 (SN-DCGAN) | STL-10 (SN-DCGAN) | ImageNet |
|---|---|---|---|
| 0.310(02) | 1.04(01) | 1.05(01) | 2.52(01) |

The error is estimated by 1std on 10 independent runs.

## Footnotes

[1] If we do not consider Wasserstein-1 but Wasserstein-2, there is no need to assume the existence of $T$ in advance and it is called Brenier's theorem [3].

[2] lr corresponds to $\epsilon$ in the main paper.

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

Figure 4: History of inception scores and FID during each transport with $lr = 0.05, k \approx 0.29$ with SNGAN model trained by CIFAR-10. Too large lr causes bad behavior.

Figure 5: History of inception scores and FID during each transport with $lr = 0.005, k \approx 0.31$ with SNGAN model trained by CIFAR-10.

Figure 6: History of inception scores and FID during each transport with $lr = 0.001, k \approx 0.28$ with SNGAN model trained by CIFAR-10. Too low lr causes slowing down the speed of improvement.