[Reviews · NeurIPS 2019]

Reviewer 1



Overall, the contributions of this paper are clear. The theoretical derivations seem correct. Also, some experiments validate the related claims in the paper.

Reviewer 2



This paper builds on ideas from Optimal transport to derive an objective for the discriminator that the authors claim is an improvement over the Wasserstein GAN. A formulation for the GAN discriminator objective is set up using Optimal Transport for 1-Wasserstein distance. This is obtained by examining transport in the Monge Problem and a proof is provided for a minimizer in this context. The Lipshitz constant K is estimated from data. The authors claim that it does not vary much, and therefore can be assumed constant. Recipes are provided for calculations in data space (Algorithm 1) and latent space (Algorithm 2). The results are quite reasonable, with examples given for images (CIFAR-10, SLT-10) and the swiss roll dataset. The paper claims that improvements are obtained over WGAN-GP (after training it for some epochs), SN-GAN. The authors' rebuttal has mostly convinced me that the formulation is sound. Most of my initial doubts had to do with the assumption that the automorphism (equation (7) in the appendix) exists, which is necessary to complete the proof that sets up the objective function. The authors draw attention to works [1] and [2] for insight. [1] L. Caffarelli, M. Feldman, and R. McCann. Journal of the American Mathematical Society, 15(1):1–26, 2002. [2] W. Gangbo and R. McCann. Acta Mathematica, 177(2):113–161, 1996. Improvements: 1) I am hoping that the language issues can be improved. 2) An explanation detailing the correctness of the assumptions pertaining to the existence of the solution to the Monge Problem (theorem 1) would be most helpful in making the paper readable. Overall: I rule this as an accept. Good paper, but the clarity could be improved. Quality: Good Clarity: Unclear. Significance: The work is highly significant in that it is of interest that we widen our theoretical and application apparatus for insight and to improve the quality of existing GAN results.

Reviewer 3



The paper makes an original contribution to the field of generative modelling that is backed by a strong theoretical analysis. It is generally written in a readable way, even though some parts could be smoothed out a little. The experiments conducted are sufficient to prove the merits of the proposed method. I think the contributions will have some significance in the field of generative modelling, even though some practical problems (i.e. runtime) are not discussed in much detail.

[Author Response · NeurIPS 2019]

We wish to express our appreciation to the reviewers for their insightful comments on our paper. In what follows, we
will answer every comment from each reviewer. All responses are reflected in our camera-ready version.

■ Reviewer #1
**> It is better to add a section: comparison with related works, to highlight the main contributions.**
Thank you for the proposal. We will add a section for comparison to DRS or more sophisticated Metropolis-Hasitings
GAN focusing on the difference between their rejection sampling-based scheme and our OT-based method.

■ Reviewer #2
**> Firstly, the quality of the writing leaves a lot to be desired for a conference such as NeurIPS. ... Secondly, I**
**felt that clarity is lacking in many places. ...**
We are sorry for that our writing makes itself hard to follow. As reviewer #2 commented, we believe fixing it will im-
prove our paper, so we will reconsider expressions from grammatical standpoint and logical viewpoint in our revise.
**> More concretely, in section 2.3, Theorem 1 is stated. ... Line 25 in supplementary material (pertaining to**
**equation (7)): "..." What is an automorphism, and what is the basis of assumption (7). To me, it looks like**
**something relating to a Monge Problem description, but I see no connection here.**
Thank you for the important comment. Yes, the automorphism $T$ we assume in Theorem 1 is the solution of cor-
responding Monge problem. We will emphasize it in our revise and explain the basis of the assumption along the
following reasoning. As commented by reviewer #3 (line 35), there is a subtle issue on uniqueness in the Monge
problem because of the norm $||x - y||_2$ that is not strictly convex. So, we need the assumption in Theorem 1 to make it
mathematically accurate because we cannot avoid using the existence of a solution in our proof. In our opinion, a proof
with another milder assumption would be too technical and out of purpose in this paper. Instead of it, let us explain
why the assumption is reasonable. One known method [1] to find a solution is based on relaxing the cost to strictly
convex cost $||x - y||_2^{1+\epsilon}$ with $\epsilon > 0$. Within the strictly convex cost, the unique solution exists [2]. So one can get an
original solution by taking limit $\epsilon \to 0$. Interestingly, their construction, $T^{(\epsilon)}(y) = \min_x\{||x - y||_2^{1+\epsilon} - D^*(x)\}$, is
almost same as ours, $T(y) = \min_x\{||x - y||_2 - D^*(x)\}$. In addition to it, DOT works only when $||x - y||_2$ is small
enough for given $y$ in our experiments. In this case, there is no big difference between $||x - y||_2$ and $||x - y||_2^{1+\epsilon}$, and
it suggests our $T$ approximates their $T^{(\epsilon)}$. All these encourage to introduce the assumption in Theorem 1.
**> A third, ... e.g equation (15) in the supplement, and equation (17) in the main paper. I believe this expression**
**might be missing a negative sign. ... In equation (15), the variable upon which integration is performed (dy) has**
**a negative sign in the delta function, suggesting that we need a negative sign. Please clarify.**
Thank you for careful reading. We think there is no need for a negative sign because the delta function is "even
function", i.e. $\delta(-x) = \delta(x)$. Rigorously speaking, it should be treated as measure and is positive by definition.

[1] L. Caffarelli, M. Feldman, and R. McCann. *Journal of the American Mathematical Society*, 15(1):1–26, 2002.
[2] W. Gangbo and R. McCann. *Acta Mathematica*, 177(2):113–161, 1996.

■ Reviewer #3
**> The proof demonstrates that T(y) is a minimizer of the right hand side of the equation, but it does not show**
**uniqueness. In fact, it is easy to construct examples ... Could the authors please comment on this and also**
**comment if this is an issue that was encountered during the experiments.**
Thank you for the important comment. It is related to reviewer #2's comment (line 12). We agree there is an issue on
uniqueness in general, like reviewer #3's example. But, there is a natural way to get a solution provided in [1] (line
32). They first relaxed the problem to $||x - y||_2^{1+\epsilon}$ by small $\epsilon > 0$, then constructed the minimizer $T^{(\epsilon)}$ ($s_\epsilon$ in their
paper, defined in line 23 in this manuscript) in essentially the same way as our derivation. $T^{(\epsilon)}$ is uniquely defined
thanks to strictly convex nature of the relaxed cost function [2] (line 33) . After that, they took $\epsilon \to 0$ limit to get the
solution. This approach is quite similar to DOT approach because $T^{(\epsilon)}$ with infinitesimally small $\epsilon$ is almost same as
$T$ in our paper by construction. Just in case, we assumed that there exists a certain map $T$ in Theorem 1 to make our
proof mathematically correct. In fact, we have not encountered any issue on experiments except for how to choose
learning rate in gradient descent DOT. And it was also discussed in lines 105∼122 in the supplementary material.
**> How does the increase in runtime of the proposed algorithm compare to using a more powerful architecture?**
We just used gradient of $G$ and $D$, so it scales same as the backprop. For reference, we put down real runtimes here.

| Tesla P100: | Swissroll (WGAN-GP) | CIFAR-10 (SN-DCGAN) | STL-10 (SN-DCGAN) | ImageNet (SN-ResNet) |
|---|---|---|---|---|
| | 0.310(02)s/30 updates | 1.04(01)s/30 updates | 1.05(01)s/30 updates | 2.52(01)s/30 updates |

The error is estimated by 1std on 10 independent runs. We will add theoretical runtime and list of the real runtimes.
**> I would have preferred the authors to use faithful reimplementations of existing architectures ... The section**
**on the experiments on ImageNet should be expanded over the CIFAR-10 section.**
We sorry for insufficient writing, but our models are based on architectures from the relevant publications, mainly
based on SNGAN paper [7] in the reference of the supplementary material. We will cite it appropriately. On the
ImageNet experiment, we will explain more including detail of the architectures. Thank you for the comment.

[Meta-Review · NeurIPS 2019]

The authors show that the discriminator in GANs increases a lower bound related to the cost of transporting from the target to the generator distribution. The approach is interesting and quite original. The quality of the writing is below average. The authors are strongly encouraged to carefully edit the paper (and perhaps seek help from someone with more experience writing technical English) before submitting the final version.